# Clinical and Neuroimaging Predictors of Alzheimer’s Dementia Conversion in Patients with Mild Cognitive Impairment Using Amyloid Positron Emission Tomography by Quantitative Analysis over 2 Years

**DOI:** 10.3390/ijerph21050547

**Published:** 2024-04-26

**Authors:** Seonjeong Kim, Daye Yoon, Junho Seong, Young Jin Jeong, Do-Young Kang, Kyung Won Park

**Affiliations:** 1Department of Neurology, Cognitive Disorders and Dementia Center, Dong-A University College of Medicine, Busan 49201, Republic of Korea; sunjung9774@gmail.com (S.K.); malvin87@naver.com (D.Y.); zunho01@naver.com (J.S.); 2Department of Nuclear Medicine, Dong-A University College of Medicine, Busan 49201, Republic of Korea; nmjeong@dau.ac.kr (Y.J.J.); dykang@dau.ac.kr (D.-Y.K.)

**Keywords:** mild cognitive impairment, Alzheimer’s dementia, amyloid positron emission tomography, ^18^F-Florapronol, posterior cingulate, clinical predictor

## Abstract

Patients with mild cognitive impairment (MCI) have a relatively high risk of developing Alzheimer’s dementia (AD), so early identification of the risk for AD conversion can lessen the socioeconomic burden. In this study, ^18^F-Florapronol, newly developed in Korea, was used for qualitative and quantitative analyses to assess amyloid positivity. We also investigated the clinical predictors of the progression from MCI to dementia over 2 years. From December 2019 to December 2022, 50 patients with MCI were recruited at a single center, and 34 patients were included finally. Based on visual analysis, 13 (38.2%) of 34 participants were amyloid-positive, and 12 (35.3%) were positive by quantitative analysis. Moreover, 6 of 34 participants (17.6%) converted to dementia after a 2-year follow-up (*p* = 0.173). Among the 15 participants who were positive for amyloid in the posterior cingulate region, 5 (33.3%) patients developed dementia (*p* = 0.066). The Clinical Dementia Rating-Sum of Boxes (CDR-SOB) at baseline was significantly associated with AD conversion in multivariate Cox regression analyses (*p* = 0.043). In conclusion, these results suggest that amyloid positivity in the posterior cingulate region and higher CDR-SOB scores at baseline can be useful predictors of AD conversion in patients with MCI.

## 1. Introduction

Mild cognitive impairment (MCI) is an early stage of cognitive dysfunction in individuals who can still perform most daily activities independently. According to previously reported results, the rate of dementia progression in healthy individuals aged >65 years is estimated to be approximately 1–2% per year but is higher at 10–15% in individuals with MCI [1]. This cognitive condition has heterogeneous features; hence, patients’ etiology and clinical course can vary. However, the most common pathology of this clinical entity is Alzheimer’s disease [2]. The concept of MCI was initially defined using clinical criteria by Petersen et al. [3]. These criteria are as follows: (a) subjective memory complaint by the patients, preferably corroborated by an informant; (b) objective memory impairment relative to age-matched and education-matched healthy people assessed by neuropsychological test; (c) normal general cognitive function; (d) normal activities of daily living; and (e) not demented. Recently, patients with MCI have been categorized into four subtypes based on memory impairment and the number of affected domains [4]. The most common subtype is amnestic mild cognitive impairment, characterized by prominent memory impairment and often considered a prodromal Alzheimer’s dementia (AD) stage [5].

Although not all patients with mild cognitive impairment will progress to dementia, they are a clinically significant group because of the increased risk for progression to AD [6]. Given the growing aging population, the number of patients with dementia is rapidly increasing, which results in significant social and economic burdens at individual and national levels. Therefore, the early identification of individuals at risk for AD conversion among those with MCI is crucial for diagnostic and therapeutic interventions [7].

Recent research has actively focused on identifying biomarkers to predict this progression, which included cerebrospinal fluid (CSF) amyloid-β 42 or the ratio of amyloid-β 42 over 40, amyloid positron emission tomography (PET), CSF total tau, threonine 181 (T181) phosphorylated tau, medial temporal atrophy on magnetic resonance imaging (MRI), and temporoparietal/precuneus hypometabolism or hypoperfusion on ^18^F-fluorodeoxyglucose (FDG)–PET [8]. Numerous advancements have been made in developing biomarkers for AD using neuroimaging approaches. These are based on pathophysiological mechanisms, such as structural decline, functional decline, connectivity decline, and pathological aggregates [9].

In particular, the pathological hallmarks of AD, such as extracellular amyloid-β plaques and intraneuronal neurofibrillary tangles consisting of hyperphosphorylated tau, have been extensively studied in postmortem Alzheimer’s dementia research. Previous studies have demonstrated that amyloid-β plaques and neurofibrillary tangles are present in varying degrees throughout the progression of dementia [10]. Notably, amyloid deposition begins many years before the clinical symptoms of dementia become apparent in patients with MCI due to Alzheimer’s pathology [11]. Even the National Aging Alzheimer’s Association revised the diagnostic criteria for Alzheimer’s disease and revealed that using amyloid biomarker tests could increase the reliability of diagnosing MCI due to Alzheimer’s disease [12]. The anterior cingulate cortex (ACC) is a part of the medial prefrontal cortex, which plays a pivotal role in memory, attention, and emotion. ACC is one of the earliest affected areas in Alzheimer’s disease and one of the most selective areas where amyloid-β accumulates at the very early stage in Alzheimer’s disease patients [13]. Therefore, amyloid PET imaging is a valuable clinical predictor for early AD detection in the predementia stage. However, few studies have reported on the predictive value of amyloid PET for clinical progression in patients with MCI.

In our study, we focused on disease progression based on clinical diagnoses that meet the criteria for probable AD as outlined by the National Institute of Neurological and Communicative Disorders and Stroke and the Alzheimer’s Disease and Related Disorders Association. To fulfill the criteria for probable AD, an affected individual must meet criterion A (the core clinical criterion) and at least one or more of the supportive biomarker criteria: B, C, D, or E. These criteria are as follows: (A) presence of early and significant episodic memory impairment; (B) presence of medial temporal lobe atrophy; (C) abnormal cerebrospinal fluid biomarkers; (D) specific pattern on functional neuroimaging with PET; and (E) proven AD autosomal dominant mutation within the immediate family [14].

This study investigated neuroimaging predictors using the newly developed amyloid tracer, 18F-Florapronol. This tracer was recently introduced by FutureChem in South Korea. 18F-Florapronol has been reported to have a structure similar to 11C-Pittsburgh compound B (PiB) and to act similarly to amyloid, and in various comparative studies with 11C-PiB PET, similar mechanisms of action and amyloid accumulation levels have been reported [15]. Some studies have shown high safety and accuracy for identifying amyloid plaques in the brain compared with 11C-PiB PET [16]. We performed visual and quantitative analyses of 18F-Florapronol amyloid PET in each brain region. Additionally, we determined conversion rates to clinically probable AD in patients with MCI based on amyloid positivity over a 2-year follow-up period.

Furthermore, other clinical predictors of progression to AD have been investigated in various studies. Longitudinal data demonstrated that the most vital factors associated with AD progression are old age and carrying at least one apolipoprotein E4 (APOE4) allele [17]. Other meta-analyses for modifiable risk factors have identified low education levels and reduced physical activity as strong predictors of AD progression [18]. Our study has longitudinally followed up on various clinical features, including neuropsychological tests and neuroimaging studies, to identify potential clinical predictors of dementia progression in patients with MCI.

## 2. Materials and Methods

### 2.1. Participants

This prospective longitudinal trial was conducted as a single-center, open-label study. Eligible participants for this study were individuals aged ≥50 years who presented with consistent cognitive complaints without significant impairment of activities of daily living. From December 2019 to December 2022, we consecutively recruited 50 patients from the cognitive disorder and dementia center at Dong-A University in South Korea. At baseline, all patients were assessed for meeting Petersen’s criteria [3]. Inclusion and exclusion criteria for the study are shown in Table 1.

### 2.2. Clinical Assessments

We recruited eligible participants for this clinical trial and collected demographic information, including sex, age, and education level. The participants underwent various tests, including physical examinations, neurologic examinations, and laboratory tests, such as complete blood count, blood chemistry test, vitamin B12/folate levels, syphilis serology, and thyroid function tests. Apolipoprotein E (APOE) genotype was determined at baseline using real-time polymerase chain reaction. Based on the number of APOE4 alleles, participants were classified as homozygotes, heterozygotes, or noncarriers.

To assess cognitive function, we conducted the Seoul Neuropsychological Screening Battery (SNSB), which includes several tests, such as the Korean version of the Mini-Mental State Examination (K-MMSE), Clinical Dementia Rating (CDR), CDR–Sum of Boxes (CDR-SOB), Global Deterioration Scale (GDS), and Korea Instrumental Activities of Daily Living (K-IADL) scale [19]. Age-, sex-, and education-specific norms derived from the results of normal controls were used to interpret the SNSB scores. Scores below the 16th percentile, corresponding to −1 SD from the norm, were considered abnormal [20]. SNSB results classified participants into three groups: amnestic single-domain MCI, amnestic multiple-domain MCI, and non-amnestic multiple-domain MCI.

They were followed up for 2 years, and their progression to AD was evaluated annually. The clinical diagnosis was made according to the criteria established by the National Institute of Neurological and Communicative Disorders and Stroke and the Alzheimer’s Disease and Related Disorders Association [14]. During the 1-year follow-up visit, assessments were conducted using K-MMSE, CDR, CDR-SOB, GDS, and K-IADL. The SNSB test was performed at the 2-year follow-up visit.

### 2.3. Neuroimaging

Brain magnetic resonance imaging was conducted using a 1.5- or 3.0-Tesla scanner. The absence of stroke, tumor, vascular malformation, traumatic brain injury, and hydrocephalus was confirmed. The white matter hyperintensities (WMHs) were assessed using a visual rating scale based on axial T2-weighted fluid-attenuated inversion recovery images. Periventricular and deep WMHs were evaluated separately, and the participants were categorized into two groups based on the presence of WMHs in either one of these regions. Medial temporal atrophy (MTA) was rated on coronal T1-weighted images using Scheltens’ visual rating scale [21]. The mean values of the left and right MTA scores were calculated, and participants were classified based on whether the mean value was ≥2.

All eligible participants underwent a brain amyloid PET scan using ^18^F-Florapronol PET. The injection dose was administered following the recommended protocol provided by the ligand manufacturer. Dynamic images were acquired 60–90 min after the intravenous injection of ^18^F-Florapronol, with a dose of 10 ± 1 mCi. Visual assessment of the PET scans was performed by a specialist trained in nuclear medicine. A scan was considered negative when the boundary between the gray and white matter was distinguishable due to lower ^18^F-Florapronol uptake in the gray matter than in the white matter. Conversely, a scan was considered positive when the ^18^F-Florapronol uptake in the gray matter was equal to or higher than that in the adjacent white matter. When comparing ^18^F-Florapronol to other radiotracers used in the health control group, ^18^F-Florapronol demonstrates less white matter accumulation, enabling better differentiation of cortical amyloid beta deposits during visual analysis. Therefore, 18F-Florapronol may offer greater advantages for visual analysis [15]. Quantitative analysis was also conducted using semi-automated quantitation method, following the procedures described in a previous report. An anatomical template was used to measure the standardized uptake value (SUV) of each brain region. The SUV ratios (SUVRs) for each region, including the frontal cortex, lateral temporal cortex, parietal cortex, occipital cortex, anterior cingulate, and posterior cingulate, were calculated using the cerebellar cortex as a reference region [22].

### 2.4. Statistical Analysis

Variables were summarized by frequency and percentage for categorical data and mean ± SD and median (range) for numeric data. Group differences were tested using the chi-squared test or Fisher’s exact test for categorical data and independent *t* test or Mann–Whitney U test for numeric data. We used the Shapiro–Wilk test to check if its distribution is normal. For data visualization, the error bar chart was used. Univariate and multivariate analyses were performed using Cox proportional hazard regression analysis to identify prognostic factors, which independently related to time to AD conversion. All statistical analyses were performed using SPSS 26.0 statistical software (IBM Corp., released 2019, IBM SPSS Statistics for Windows, Version 26.0, Armonk, NY, USA), and *p* values of <0.05 were considered statistically significant.

## 3. Results

We initially recruited 50 patients for the study, but 11 patients were excluded as they did not meet the diagnostic criteria for MCI (Figure 1). The enrolled 39 patients were followed up for 1 year, but 5 patients were lost to follow-up and dropped out within 2 years. Consequently, the final analysis included 34 participants. The visual analysis of ^18^F-Florapronol PET revealed increased amyloid depositions in 13 of 34 (38.2%) participants. Meanwhile, a quantitative analysis showed positive amyloid deposits in 12 of 34 (35.3%) participants.

The visual analysis compared the two groups for amyloid positivity (Table 2). The male-to-female ratio was 12:22, and the mean age was 70.77 ± 6.14 years, with a mean education duration of 10.12 ± 3.60 years. Among the 27 participants who underwent the APOE genotype test, 1 (3.7%) was homozygous with two APOE4 alleles, 5 (18.5%) participants were heterozygous with one APOE4 allele, and 21 (77.8%) participants were noncarriers. In terms of the MCI subtype, 26 (76.5%) of 34 participants belonged to an amnestic multiple MCI, whereas 8 (23.5%) participants belonged to non-amnestic multiple MCI.

Throughout the study, four (11.8%) patients converted to AD at 1 year, and six (17.6%) patients converted to AD at the 2-year follow-up. In the quantitative analysis, we classified the participants into two groups based on amyloid positivity using SUVRs for each brain region and a cut-off value. Significant correlations were observed when comparing the two groups based on amyloid positivity determined by visual and quantitative analyses.

The basic demographics and clinical characteristics did not differ significantly between the two groups, except for the K-IADL at baseline (*p* = 0.004) and 1-year follow-up (*p* = 0.001), which show higher scores. Moreover, at the 2-year follow-up, the amyloid-positive group exhibited lower scores on the K-MMSE (*p* = 0.047) and higher GDS (*p* = 0.029) scales.

In our study, 6 (17.6%) of 34 participants were converted to AD. We mainly investigated the conversion rate to AD among the amyloid-positive group at 2 years (Table 3). As a result, 4 (30.8%) of 13 participants identified as amyloid-positive by visual analysis, and 4 (33.3%) of 12 participants identified as amyloid-positive by quantitative analysis progressed to AD. However, no significant association was found between amyloid positivity and AD conversion. As is well established, the reduced cortical thickness of the rostral ACC can predict the conversion from amnestic MCI to AD with psychosis. This finding underscores the potential importance of the ACC in the pathogenesis of Alzheimer’s disease [23]. Meanwhile, 5 (33.3%) of 15 participants with amyloid positivity in the posterior cingulate region progressed to AD when evaluating each brain region. This finding exhibited marginal significance (*p* = 0.066), and post hoc power was estimated to be approximately 56% based on the observed effect size in our study.

To identify predictors affecting the progression to AD, we evaluated the clinical characteristics by comparing the remained MCI group with the AD conversion group (Table 4). Although the basic demographics were not significantly different between the two groups, the AD conversion group had a higher CDR-SOB score at baseline (*p* = 0.002). At 1-year follow-up, the AD conversion group showed lower scores of K-MMSE (*p* = 0.022) and higher scores of CDR-SOB (*p* = 0.001) and GDS (*p* = 0.003). Similar results were observed at the 2-year follow-up, such as lower K-MMSE (*p* = 0.005), higher CDR-SOB (*p* = 0.002), and GDS (*p* = 0.000) scales.

We performed the univariate and multivariate Cox regression analyses to examine the effect of the independent variable on time to AD conversion. The statistically significant variables were selected in a backward elimination method with an α level of 0.05. We conducted two models of multivariate analysis using Cox regression analysis. In the first model, the data were considered for all sociodemographic and clinical variables at baseline and the first assessment scores of K-MMSE, CDR, CDR-SOB, GDS, and K-IADL. We added 1- and 2-year follow-up scores as covariates in the second model (Table 5). The result of analyzing the correlation between Amyloid-ß composite SUVRs and CDR-SOB by Pearson’s correlation test shows that 1-year (r = 0.3440, *p* > 0.05) and 2-year follow-up CDR-SOB (r = 0.3022, *p* > 0.05) have a weak correlation (Figure 2). CDR-SOB at baseline showed a significant association with conversion to dementia (hazard ratio [HR] = 3.757; 95% confidence interval [95% CI], 1.041–13.556, *p* = 0.043). Additionally, K-MMSE at 1 year (HR = 0.629; 95% CI, 0.395–1.001, *p* = 0.051) and K-IADL at 2 years (HR = 8.069; 95% CI, 0.997–65.311, *p* = 0.050) were selected as variables associated with increased risk of AD conversion. Therefore, higher CDR-SOB scores at baseline, lower K-MMSE scores at the 1-year follow-up, and higher K-IADL scores at the 2-year follow-up were significantly related to an increased risk of AD conversion.

## 4. Discussion

A critical goal of biomedical research is establishing early AD indicators during the preclinical stage for diagnosis and intervention. Remarkable advances have recently been made in diagnostic techniques for AD, including developing several PET imaging agents. Amyloid PET tracers enable quantitative measurement of the insoluble cortical Aβ load in vivo. 11C-PIB was the first agent developed for detecting cerebral Aβ accumulation [24]. Despite the high sensitivity and specificity of the 11C-PIB tracer, it is not commonly used in clinical practice due to its short half-life. Currently, clinically available amyloid PET tracers were ^18^F-Florbetapir, ^18^F-Flutemetamol, ^18^F-Florbetaben, and ^18^F-Florapronol, which have an extended half-life. Among them, studies using ^18^F-Florbetapir have been relatively actively conducted. In the longitudinal study using ^18^F-Florbetapir, the frequency of amyloid positivity was 196 of 401 (48.9%) by visual analysis and 221 (55.1%) of 401 by quantitative analysis. Although visual analysis has a lower sensitivity and higher specificity than quantitative analysis, two methods may be used to determine amyloid positivity in patients with MCI. Furthermore, the conversion rate was 15.2% within a mean of 1.6 years, and a positive scan of ^18^F-Florbetapir PET was associated with a greater hazard for AD conversion, which was affected by APOE4 and baseline cognitive status [25].

Only a few studies have used the newly developed amyloid tracer ^18^F-Florapronol. In our study, 38.2% (13 of 34) participants showed increased amyloid deposition based on visual analysis of brain amyloid PET using an ^18^F-Florapronol tracer, whereas 35.3% (12 of 34) participants showed increased amyloid deposition based on quantitative analysis. These findings are consistent with previous studies showing that amyloid deposition occurs early in AD, even before clinical symptoms manifest [26].

The difference between the two methods arises because quantitative analysis was performed using composite SUVRs, meaning that even partial positivity observed visually may be diluted and confirmed as negative when assessed quantitatively using average values. Additionally, the comparison of amyloid accumulation between white matter and cortex was conducted for qualitative analysis, while quantitative analysis used the cerebellum as the reference region [27].

Moreover, there was a high concordance between visual and quantitative analyses in determining amyloid positivity using ^18^F-Florapronol PET. Therefore, we demonstrated that ^18^F-Florapronolas, as an amyloid tracer for detecting amyloid pathology in patients with MCI, is a promising tool.

We further explored the association between amyloid positivity and conversion to AD. AD conversion rates within the amyloid-positive group were 30.8% (4 of 13) participants based on visual analysis and 33.3% (4 of 12) participants based on quantitative analysis. Previous studies have investigated the predictive value of amyloid PET in patients with MCI. In the 11C-PiB PET study, 17 (55%) of 31 patients with MCI had increased 11C-PiB retention at baseline and 14 (82%) of 17 patients clinically converted to AD. The longitudinal study over 2 years found faster converters within 1 year of baseline PET had higher tracer retention in the anterior cingulate and frontal cortex [28] 양식의 맨 위. However, our results indicate that amyloid positivity in MCI patients was not significantly associated with the conversion to AD over a 2-year follow-up period. These findings suggest that amyloid deposition alone is not sufficient to predict AD conversion in mild cognitive impairment patients.

It is noteworthy that a marginal significance was found in the association between amyloid positivity in the posterior cingulate and AD conversion when evaluating specific brain regions using quantitative analysis. This suggests that amyloid deposition in the posterior cingulate has a stronger predictive value for progression to AD in patients with MCI.

A major limitation of amyloid imaging and studies of amyloid burden, generally, is a poorly understood relationship with cognition. Clinicopathologic studies have revealed that cognition and clinical state are more closely related to the distribution and burden of neurofibrillary tangles than the topography or degree of amyloid plaque. Most recently, it has been shown that amyloid deposition predicts tau deposition in aging, suggesting that amyloid binding varies on a continuum [29]. Notably, parietal cortices, such as the posterior cingulate, retrosplenial cortex, and precuneus, are heavily interconnected with the medial temporal lobe, which are sites for early aggregation of tau pathology [30].

When comparing the remaining MCI and AD conversion groups, we observed minimal differences in sex, age, education level, and APOE subtype. No participants belonged to single amnestic MCI in our study, and there were more patients with multiple amnestic MCI than nonamnestic multiple MCI in the AD conversion group. Regarding the clinical characteristics, the baseline CDR-SOB score was significantly higher in the AD conversion group than in the remained MCI group. These findings indicate that greater cognitive impairment and functional decline are associated with an increased risk of conversion to AD. Additionally, longitudinal assessments at the 1- and 2-year follow-up visits revealed lower K-MMSE and higher CDR-SOB and GDS scores in the AD conversion group than the remaining MCI group. These results support the notion that worsening cognitive and functional ability decline over time indicates disease progression.

Meanwhile, white matter hyperintensities (WMHs) and medial temporal atrophy (MTA) did not show significant differences between the remaining MCI group and the AD conversion group. This suggests that WMHs and MTA are not possible as reliable predictors for AD conversion in patients with MCI, at least within the context of this study.

In the multivariate Cox regression analysis, higher CDR-SOB scores at baseline, lower K-MMSE scores at 1-year follow-up, and higher K-IADL scores at 2-year follow-up were significantly associated with an increased risk for conversion to AD in patients with MCI. Mainly predicting the risk of progression at the early stage of the disease holds a greater value; therefore, our results highlight the significant potential of the baseline CDR-SOB scores as a predictor for conversion to AD in patients with MCI.

The diagnostic impact and clinical availability of amyloid PET are still under the surface. Most clinically diagnosed patients with AD show amyloid PET-positive findings, but the age of onset and the presence of the APOE4 allele affect the amyloid positivity rate [31]. Although a negative scan on amyloid PET can almost rule out AD, in one meta-analysis, the prevalence of a positive scan was 88% in patients with AD, 51% in patients with dementia with Lewy bodies, 12% in patients with frontotemporal dementia, 38% in patients with corticobasal degeneration, and 24% in healthy individuals [32]. Although Alzheimer’s pathology is not a direct cause of Alzheimer’s disease, it has been established as a pathological finding that can differentiate AD from other degenerative dementias that can cause dementia [33]. In addition, amyloid PET has been an informative tool in AD biomarker research for staging disease progression and selecting individuals for participation in biomarker-based clinical trials during the asymptomatic phase [34].

Our study had several limitations. First, the sample size was relatively small, which may have limited the statistical power to detect significant associations. Primarily, the result of the association between posterior cingulate amyloid positivity and the risk for AD conversion can be more reliable if further study is conducted with a large cohort. Second, the follow-up period of 2 years may not be sufficient to capture all cases of AD conversion because the rate of progression from MCI to AD can vary; therefore, long-term follow-up studies are needed to validate our findings. Third, the study focused on a specific amyloid tracer, which may limit the generalization of the results. Although we performed visual and quantitative analyses of ^18^F-Florapronol amyloid PET, the accuracy and reliability of this tracer compared with other amyloid PET tracers have not been extensively studied. Finally, the analysis did not include other potential predictors, such as genetic factors and biomarkers beyond amyloid deposition. Further research should incorporate a broader range of predictors to enhance predictive accuracy.

## 5. Conclusions

This study provides insights into the clinical and neuroimaging predictors for converting to AD in patients with MCI. The results suggest that although amyloid deposition alone is not a strong predictor, amyloid deposition in the posterior cingulate has potential predictive value. The use of ^18^F-Florapronol PET and comprehensive clinical assessments provides valuable information for identifying individuals at risk of developing AD. Additionally, baseline cognitive and functional impairment could be clinical predictors of disease progression. One of the key findings of this study is the importance of baseline functional impairment assessed by the CDR-SOB score, which could predict the conversion from MCI to AD. These findings contribute to understanding AD progression in patients with MCI and may have implications for early detection and intervention strategies. However, it is difficult to conclude that CDR-SOB is a predictive neurocognitive biomarker in the predementia stage due to the small sample size. Further research is warranted to validate and expand upon these findings.

## Figures and Tables

**Figure 1 ijerph-21-00547-f001:**
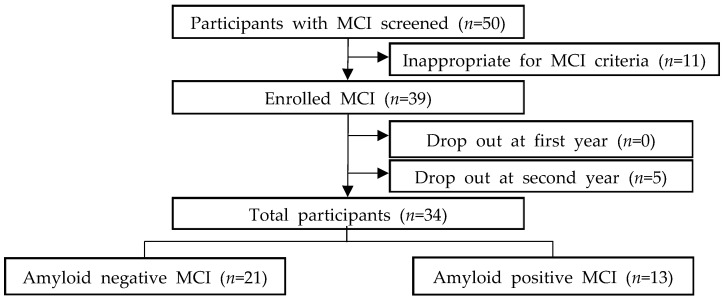
Flowchart of the participants. MCI, mild cognitive impairment.

**Figure 2 ijerph-21-00547-f002:**
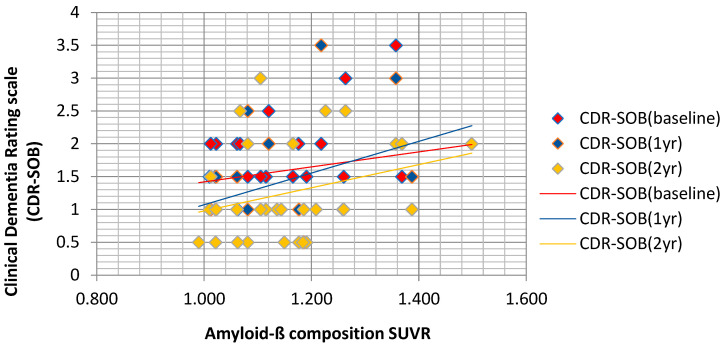
Correlation plot between CDR-SOB and Amyloid-ß composite SUVRs. CDR-SOB, Clinical. Dementia Rating–Sum of Boxes; SUVRs, standardized uptake value ratios.

**Table 1 ijerph-21-00547-t001:** Inclusion and exclusion criteria for a prospective longitudinal study of patients with mild cognitive impairment (MCI).

Inclusion Criteria
∙ The presence of consistent complaints of cognitive decline reported by either the patient or the caregiver
∙ Lower scores of one or more cognitive domains based on neuropsychological battery below the average of −1 standard deviation (SD) of normal considering age and education
∙ Clinical Dementia Rating (CDR) score of 0.5, with a score of 0.5 or 1 on the memory item
∙ No significant impairment in activities of daily living
∙ Not diagnosed with dementia by clinician
∙ Modified Hachinski ischemic score (HIS) of ≤4
∙ The ability to read and write
∙ Age range of 50–90 years
∙ Stable use of acetylcholinesterase inhibitors or N-methyl-D-aspartate (NMDA) receptor antagonist for at least 8 weeks prior to consent or no use of these drugs
∙ The absence of any brain lesions (e.g., tumor, stroke, or subdural hematoma) that can potentially cause cognitive impairment
∙ Written informed consent provided by the participants
Exclusion Criteria
∙ Severe physical illnesses that could interfere with the clinical study
∙ Major psychiatric disorders that would hinder the performance of amyloid PET
∙ The presence of other neurologic disorders
∙ Recent heart surgery or diagnosis of myocardial infarction within 6 months before screening
∙ Scheduled to receive radiopharmaceuticals for treatment or participation in other clinical trials that may affect PET image acquisition considering effective half-life
∙ Pregnancy, lactation, or premenopausal women planning a pregnancy
∙ Assessed as unsuitable for participation in the clinical trial

**Table 2 ijerph-21-00547-t002:** Demographics data and clinical characteristics between the two groups regarding amyloid positivity by visual analysis at baseline and follow-up.

Variables	Overall (*n* = 34)	Amyloid Negative (*n* = 21)	Amyloid Positive (*n* = 13)	*p*-Value
Sex (female)	22 (64.7)	12 (57.1)	10 (76.9)	0.292 ^4^
Age (year)	70.77 ± 6.14	71.86 ± 6.53	69.00 ± 5.20	0.191 ^1^
Education (years)	10.12 ± 3.60	10.67 ± 3.75	9.23 ± 3.30	0.267 ^2^
APOE4 genotyping				
Noncarrier	21 (77.8)	15 (88.2)	6 (60.0)	0.176 ^3^
Heterozygote	5 (18.5)	2 (11.8)	3 (30.0)	
Homozygote	1 (3.7)	0 (0.0)	1 (10.0)	
White matter hyperintensities	4 (14.3)	3 (18.8)	1 (8.3)	0.613 ^4^
Medial temporal atrophy	11 (39.3)	8 (50.0)	3 (25.0)	0.253 ^4^
MCI subtype				
Amnestic multiple	26 (76.5)	14 (66.7)	12 (92.3)	0.116 ^4^
Nonamnestic multiple	8 (23.5)	7 (33.3)	1 (7.7)	
AD conversion in first year	4 (11.8)	1 (4.8)	3 (23.1)	0.274 ^4^
AD conversion in second year	6 (17.6)	2 (9.5)	4 (30.8)	0.173 ^4^
Amyloid positive by quantitative analysis (SUVR cutoff value)				
Composite (≥1.186)	12 (35.3)	2 (9.5)	10 (76.9)	0.000 ^4^
Frontal lobe (≥1.094)	16 (47.1)	5 (23.8)	11 (84.6)	0.001 ^4^
Lateral temporal lobe (≥1.254)	12 (35.3)	2 (9.5)	10 (76.9)	0.000 ^4^
Parietal lobe (≥1.154)	13 (38.2)	3 (14.3)	10 (76.9)	0.001 ^4^
Anterior cingulate (≥1.368)	10 (29.4)	3 (14.3)	7 (53.8)	0.022 ^4^
Posterior cingulate (≥1.380)	15 (44.1)	6 (28.6)	9 (69.2)	0.034 ^4^
K-MMSE (baseline)	25.94 ± 2.72	26.57 ± 1.45	24.92 ± 3.50	0.086 ^1^
CDR (baseline)	0.50 ± 0.00	0.5 ± 0.00	0.5 ± 0.00	1.000 ^2^
CDR-SOB (baseline)	1.60 ± 0.76	1.50 ± 0.74	1.77 ± 0.78	0.385 ^2^
GDS (baseline)	3.03 ± 0.30	3.00 ± 0.32	3.08 ± 0.28	0.471 ^2^
K-IADL (baseline)	0.10 ± 0.16	0.08 ± 0.19	0.14 ± 0.08	0.004 ^2^
K-MMSE (1 year)	25.27 ± 2.91	25.62 ± 2.60	24.70 ± 3.38	0.374 ^1^
CDR (1 year)	0.50 ± 0.00	0.50 ± 0.00	0.50 ± 0.00	1.000 ^2^
CDR-SOB (1 year)	1.46 ± 0.85	1.24 ± 0.75	1.81 ± 0.90	0.058 ^2^
GDS (1 year)	3.09 ± 0.38	3.05 ± 0.38	3.15 ± 0.38	0.437 ^2^
K-IADL (1 year)	0.13 ± 0.16	0.06 ± 0.08	0.24 ± 0.21	0.001 ^2^
K-MMSE (2 years)	24.53 ± 3.03	25.33 ± 2.48	23.23 ± 3.47	0.047 ^1^
CDR (2 years)	0.50 ± 0.00	0.50 ± 0.00	0.50 ± 0.00	1.000 ^2^
CDR-SOB (2 years)	1.37 ± 0.81	1.17 ± 0.73	1.70 ± 0.85	0.055 ^2^
GDS (2 years)	3.15 ± 0.50	3.00 ± 0.45	3.39 ± 0.51	0.029 ^2^
K-IADL (2 years)	0.21 ± 0.36	0.13 ± 0.25	0.34 ± 0.47	0.084 ^2^

Data are presented as mean ± standard deviation (SD) or number (%) unless otherwise indicated. Shapiro–Wilk test was used for the test of the normality assumption. AD, Alzheimer’s dementia; APOE, apolipoprotein E; CDR, Clinical Dementia Rating; CDR-SOB, CDR–Sum of Boxes; GDS, Global Deterioration Scale; K-IADL, Korea Instrumental Activities of Daily Living scale; K-MMSE, the Korean version of the Mini-Mental State Examination; MCI, mild cognitive impairment; SUVR, standardized uptake value ratio. ^1^ *p*-values were derived from an independent *t* test. ^2^ *p*-values were derived from Mann–Whitney’s U test. ^3^ *p*-values were derived from the chi-squared test. ^4^ *p*-values were derived from Fisher’s exact test.

**Table 3 ijerph-21-00547-t003:** Comparison of Alzheimer’s dementia conversion between groups according to amyloid positivity at 2 years.

Variables	Remained MCI (*n* = 28)	AD Conversion at 2 Years (*n* = 6)	*p*-Value
Amyloid positivity by visual analysis			
Negative	19 (90.5)	2 (9.5)	0.173
Positive	9 (69.2)	4 (30.8)	
Amyloid positivity by quantitative analysis			
Negative	20 (90.9)	2 (9.1)	0.154
Positive	8 (66.7)	4 (33.3)	
Composite			
<1.186	20 (90.9)	2 (9.1)	0.154
≥1.186	8 (66.7)	4 (33.3)	
Frontal lobe			
<1.094	16 (88.9)	2 (11.1)	0.387
≥1.094	12 (75.0)	4 (25.0)	
Lateral temporal lobe			
<1.254	19 (88.4)	3 (13.6)	0.641
≥1.254	9 (75.0)	3 (25.0)	
Parietal lobe			
<1.154	19 (90.5)	2 (9.5)	0.173
≥1.154	9 (69.2)	4 (30.8)	
Occipital lobe			
<1.245	16 (94.1)	1 (5.9)	0.175
≥1.245	12 (70.6)	5 (29.4)	
Anterior cingulate			
<1.368	21 (87.5)	3 (12.5)	0.328
≥1.368	7 (70.0)	3 (30.0)	
Posterior cingulate			
<1.380	18 (94.7)	1 (5.3)	0.066
≥1.380	10 (66.7)	5 (33.3)	

*p*-value was derived from Fisher’s exact test. AD, Alzheimer’s dementia; MCI, mild cognitive impairment.

**Table 4 ijerph-21-00547-t004:** Comparison of clinical characteristics between groups regarding Alzheimer’s dementia conversion at 2 years.

Variables	Overall (*n* = 34)	Remained MCI (*n* = 28)	AD Conversion at 2 Years (*n* = 6)	*p*-Value
Sex (female)	22 (64.7)	19 (67.9)	3 (50.0)	0.641 ^4^
Age (year)	70.77 ± 6.14	70.86 ± 6.28	70.33 ± 5.92	0.853 ^1^
Education (years)	10.12 ± 3.60	10.25 ± 3.60	9.50 ± 3.89	0.643 ^2^
APOE4 genotyping				
Noncarrier	21 (77.8)	18 (78.3)	3 (75.0)	0.867 ^3^
Heterozygote	5 (18.5)	4 (17.4)	1 (25.0)	
Homozygote	1 (3.7)	1 (4.3)	0 (0.0)	
White matter hyperintensities	4 (14.3)	3 (13.0)	1 (20.0)	1.000 ^4^
Medial temporal atrophy	11(39.3)	9(39.1)	2(40.0)	1.000 ^4^
MCI subtype				
Amnestic multiple	26 (76.5)	21 (75.0)	5 (83.3)	1.000 ^4^
Nonamnestic multiple	8 (23.5)	7 (25.0)	1 (16.7)	
K-MMSE (baseline)	25.94 ± 2.72	26.32 ± 2.23	24.17 ± 4.17	0.268 ^1^
CDR (baseline)	0.50 ± 0.00	0.50 ± 0.00	0.50 ± 0.00	1.000 ^2^
CDR-SOB (baseline)	1.60 ± 0.76	1.43 ± 0.65	2.42 ± 0.74	0.002 ^1^
GDS (baseline)	3.03 ± 0.30	3.0 ± 0.27	3.17 ± 0.41	0.215 ^2^
K-IADL (baseline)	0.10 ± 0.16	0.09 ± 0.16	0.17 ± 0.15	0.081 ^2^
K-MMSE (1 year)	25.27 ± 2.91	25.79 ± 2.71	22.83 ± 2.71	0.022 ^2^
CDR (1 year)	0.50 ± 0.00	0.50 ± 0.00	0.50 ± 0.00	1.000 ^2^
CDR-SOB (1 year)	1.46 ± 0.85	1.21 ± 0.67	2.58 ± 0.66	0.001 ^2^
GDS (1 year)	3.09 ± 0.38	3.00 ± 0.27	3.50 ± 0.55	0.003 ^2^
K-IADL (1 year)	0.13 ± 0.16	0.11 ± 0.12	0.27 ± 0.28	0.069 ^2^
K-MMSE (2 years)	24.53 ± 3.03	25.18 ± 2.74	21.5 ± 2.59	0.005 ^1^
CDR (2 years)	0.50 ± 0.00	0.50 ± 0.00	0.50 ± 0.00	1.000 ^2^
CDR-SOB (2 years)	1.37 ± 0.81	1.14 ± 0.65	2.42 ± 0.66	0.002 ^2^
GDS (2 years)	3.15 ± 0.50	3.00 ± 0.38	3.83 ± 0.41	0.000 ^2^
K-IADL (2 years)	0.21 ± 0.36	0.12 ± 0.20	0.62 ± 0.62	0.0801

Unless otherwise indicated, data are presented as mean ± standard deviation (SD) or number (%). Shapiro–Wilk test was used for the test of the normality assumption. AD, Alzheimer’s dementia; APOE, apolipoprotein E; CDR, Clinical Dementia Rating; CDR-SOB, CDR–Sum of Boxes; GDS, Global Deterioration Scale; K-IADL, Korea Instrumental Activities of Daily Living scale; K-MMSE, the Korean version of the Mini-Mental State Examination; MCI, mild cognitive impairment. ^1^ *p*-values were derived from an independent *t* test. ^2^ *p*-values were derived from Mann–Whitney’s U test. ^3^ *p*-values were derived from the chi-square test. ^4^ *p*-values were derived from Fisher’s exact test.

**Table 5 ijerph-21-00547-t005:** Univariate and multivariate Cox proportional hazard regression analyses for time to Alzheimer’s dementia conversion at 2 years.

Variables	Univariate Analysis			Multivariate Analysis		
	HR	95% CI	*p*-Value	HR	95% CI	*p*-Value
Sex (female)	0.530	(0.107–2.626)	0.437			
Age (year)	0.984	(0.862–1.124)	0.814			
Education (years)	0.941	(0.744–1.189)	0.610			
APOE4 genotyping						
Noncarrier	1.000					
Heterozygote	8046.5	(0.000–24,541)	0.969			
Homozygote	12,267.8	(0.000–37,528)	0.968			
White matter hyperintensities	1.545	(0.173–13.834)	0.697			
Medial temporal atrophy	1.046	(0.175–6.260)	0.961			
MCI subtype						
Amnestic multiple	1.000					
Nonamnestic multiple	1.630	(0.190–13.965)	0.656			
Amyloid positive by visual analysis						
Amyloid positive	3.475	(0.635–19.009)	0.151			
Amyloid positive by quantitative analysis (SUVR cutoff value)						
Composite (≥1.186)	3.982	(0.728–21.794)	0.111			
Frontal lobe (≥1.094)	2.368	(0.433–12.939)	0.320			
Lateral temporal lobe (≥1.254)	1.887	(0.381–9.354)	0.437			
Parietal lobe (≥1.154)	3.475	(0.635–19.009)	0.151			
Occipital lobe (≥1.245)	5.235	(0.611–44.832)	0.131			
Anterior cingulate (≥1.368)	2.522	(0.506–12.453)	0.260			
Posterior cingulate (≥1.380)	6.715	(0.784–57.531)	0.082			
K-MMSE (baseline)	0.798	(0.624–1.021)	0.073			
CDR (baseline)	-	-	-			
CDR-SOB (baseline)	4.095	(1.441–11.643)	0.008	3.757	(1.041–13.556)	0.043
GDS (baseline)	3.903	(0.500–30.490)	0.194			
K-IADL (baseline)	5.290	(0.179–156.10)	0.335			
K-MMSE (1 year)	0.712	(0.519–0.975)	0.034	0.629	(0.395–1.001)	0.051
CDR (1 year)	-	-	-			
CDR-SOB (1 year)	5.376	(1.760–16.419)	0.003			
GDS (1 year)	9.013	(1.811–44.856)	0.007			
K-IADL (1 year)	30.567	(1.299–719.04)	0.034			
K-MMSE (2 years)	0.668	(0.487–0.917)	0.013			
CDR (2 years)	-	-	-			
CDR-SOB (2 years)	4.415	(1.656–11.767)	0.003			
GDS (2 years)	21.839	(2.551–186.98)	0.005			
K-IADL (2 years)	5.542	(1.431–21.462)	0.013	8.069	(0.997–65.311)	0.050

The effect of independent variables on time to AD conversion was analyzed using the multivariate Cox regression, and the statistically significant variables were selected in a backward elimination method with 0.05 alpha level. HR, hazard ratio; CI, confidence interval; APOE, apolipoprotein E; K-MMSE, Korean version of the Mini-Mental State Examination; CDR, Clinical Dementia Rating; CDR-SOB, CDR–Sum of Boxes; GDS, Global Deterioration Scale; K-IADL, Korea Instrumental Activities of Daily Living scale.

## Data Availability

The data are not publicly available due to patient confidentiality and privacy.

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
