# Peer review of "Clinical and Neuroimaging Predictors of Alzheimer’s Dementia Conversion in Patients with Mild Cognitive Impairment Using Amyloid Positron Emission Tomography by Quantitative Analysis over 2 Years"

_ijerph, 2024, doi:10.3390/ijerph21050547_

Round 1
Reviewer 1 Report
Comments and Suggestions for Authors
Basic reporting
- This paper examines the association between beta-amyloid deposition and clinical progression to AD dementia
- The paper reports the following major findings:
1. Positive findings are:
a) Clinical progression to AD depends on amyloid deposition in the posterior cingulate gyrus.
b) Clinical progression to AD depends on a higher CDR-sum of boxes at baseline ( entry point in the study.
2. Negative findings are:
a) Clinical progression to AD is not dependent on global beta-amyloid deposition.
The paper clearly conveys the literature, methods, and major findings. There are minor grammatical errors, for example, in line 67 “and’ should be omitted. Proofreading can easily correct these minor mistakes.
Overall, the paper is self-contained with relevant results to support the claims.
Experimental design
Strengths of the paper
1. The paper includes clearly describes the introduction, methods, and major findings.
2. The paper performs sound statistical analysis and reports the details enough to replicate the study.
3. Conclusions are supported by the evidence presented in the paper.
4. The paper correctly identifies all major limitations of the study.
Weaknesses
1. Line 53 mentions CSF beta-amyloid 42 as a marker, however, per Jack et al 2018 the ratio of beta-amyloid 42 over 40 is a marker of beta-amyloid in the brain.
2. The authors discuss the progression to AD however, do not clarify which definition of AD they are utilizing. In 2018 definition (by Jack et al) other markers would be required such as tau and neurodegeneration. Otherwise, the paper would have to specify that the patients were on the AD continuum (when only beta-amyloid status is known.
2. It would be helpful to have a potential explanation of why there were differences in beta-amyloid characterization qualitatively and quantitatively. Were these the same 12 patients in both groups (qualitative and quantitative) with an additional patient in quantitative.
3. It would be suggested to describe why white matter and gray matter comparison was done for beta-amyloid deposition. Although the white matter has lower beta-amyloid still for PIB cerebellum is still used as a comparison instead of white matter.
Comments on the Quality of English LanguageQuality of English is good with minor corrections on proof reading.
Reviewer 2 Report
Comments and Suggestions for Authors
Well written paper.
interesting topic.
I have only one question
Amyloid Positivity by Visual Analysis are very similar to Amyloid Positivity by Quantitative Analysis.
what is the best?
Author Response
1. Amyloid Positivity by Visual Analysis are very similar to Amyloid Positivity by Quantitative Analysis.
what is the best?
Thank you for your question. The golden standard for amyloid beta analysis is visual analysis, while quantitative analysis is commonly conducted in group studies for convenience. Studies utilizing the ADNI (Alzheimer's Disease Neuroimaging Initiative) dataset are prominent in this regard.
References
Petersen RC, Aisen PS, Beckett LA, Donohue MC, Gamst AC, Harvey DJ, Jack CR Jr, Jagust WJ, Shaw LM, Toga AW, Trojanowski JQ, Weiner MW. Alzheimer's Disease Neuroimaging Initiative (ADNI): clinical characterization. Neurology. 2010 Jan 19;74(3):201-9.
Reviewer 3 Report
Comments and Suggestions for Authors
The authors presented “Clinical and Neuroimaging Predictors of Alzheimer’s Dementia Conversion in Patients with Mild Cognitive Impairment Using Amyloid Positron Emission Tomography by Quantitative Analysis over 2 Years." Overall, the findings are interesting. There are several comments that need to address.
Strength of the study: Study is important as it implicates CDR-SOB crucial to predict shift from MCI to dementia.
Limitation of the study: Small sample size.
Abstract:
· Authors must avoid using too many abbreviations.
Introduction:
· Better to define all the abbreviations throughout the manuscript and minimize its use.
· Elaboration on how 18F-Florapronol interact with amyloid beta and its comparison other existing tracer would improve quality of manuscript.
· Author must elaborate importance of anterior cingulate in MCI and dementia.
Methodology:
· I would appreciate if authors can provide inclusion and exclusion criteria in table form.
· The sample size is small, and it would be good if the author could add physical function performance and its data in result: if they have it.
Results:
· Please do the correlation plot between CDR-SOB score and amyloid beta.
· Results are short and I would appreciate if they elaborated with the importance fore example what they observed in anterior cingulate and why it is important compared to prefrontal cortex. Author can use reference for it.
Discussion and conclusion:
· In line no 350 authors have used term tauopathy and I would recommend being specific with amyloid beta and anterior cingulate.
· Discussion seems to be elaborated form of result. I would recommend that authors must discuss their findings with the existing evidences.
· It hard to conclude CDR-SOB predictive neurocognitive biomarker in the predementia stage as the sample size is less. Please modify the sentence.
